# A Novel Nutrient- and Antioxidant-Based Formulation Can Sustain Tomato Production under Full Watering and Drought Stress in Saline Soil

**DOI:** 10.3390/plants12193407

**Published:** 2023-09-27

**Authors:** Taia A. Abd El-Mageed, Radwa Ihab, Mostafa M. Rady, Hussein E. E. Belal, Fatma A. Mostafa, Tarek M. Galal, Luluah M. Al Masoudi, Esmat F. Ali, Maria Roulia, Amr E. M. Mahmoud

**Affiliations:** 1Soil and Water Science Department, Faculty of Agriculture, Fayoum University, Fayoum 63514, Egypt; taa00@fayoum.edu.eg; 2Botany Department, Faculty of Agriculture, Fayoum University, Fayoum 63514, Egypt; radwaihab4@gmail.com (R.I.); hes00@fayoum.edu.eg (H.E.E.B.); 3Plant Pathology Research Institute, Agriculture Research Center, Giza 11571, Egypt; drfatmamostafa54@gmail.com; 4Department of Biology, College of Science, Taif University, P.O. Box 11099, Taif 21944, Saudi Arabia; t.aboseree@tu.edu.sa (T.M.G.); lm.al-masoudi@tu.edu.sa (L.M.A.M.); 5Inorganic Chemistry Laboratory, Department of Chemistry, National and Kapodistrian University of Athens, 157 72 Athens, Greece; 6Biochemistry Department, Faculty of Agriculture, Fayoum University, Fayoum 63514, Egypt; aem01@fayoum.edu.eg

**Keywords:** antioxidants, drought and salt stress, leaf integrity, tomato, yield

## Abstract

As a result of the climate changes that are getting worse nowadays, drought stress (DS) is a major obstacle during crop life stages, which ultimately reduces tomato crop yields. So, there is a need to adopt modern approaches like a novel nutrient- and antioxidant-based formulation (NABF) for boosting tomato crop productivity. NABF consists of antioxidants (i.e., citric acid, salicylic acid, ascorbic acid, glutathione, and EDTA) and nutrients making it a fruitful growth stimulator against environmental stressors. As a first report, this study was scheduled to investigate the foliar application of NABF on growth and production traits, physio-biochemical attributes, water use efficiency (WUE), and nutritional, hormonal, and antioxidative status of tomato plants cultivated under full watering (100% of ETc) and DS (80 or 60% of ETc). Stressed tomato plants treated with NABF had higher DS tolerance through improved traits of photosynthetic efficiency, leaf integrity, various nutrients (i.e., copper, zinc, manganese, calcium, potassium, phosphorus, and nitrogen), and hormonal contents. These positives were a result of lower levels of oxidative stress biomarkers as a result of enhanced osmoprotectants (soluble sugars, proline, and soluble protein), and non-enzymatic and enzymatic antioxidant activities. Growth, yield, and fruit quality traits, as well as WUE, were improved. Full watering with application of 2.5 g NABF L^−1^ collected 121 t tomato fruits per hectare as the best treatment. Under moderate DS (80% of ETc), NABF application increased fruit yield by 10.3%, while, under severe DS (40% of ETc), the same fruit yield was obtained compared to full irrigation without NABF. Therefore, the application of 60% ETc × NABF was explored to not only give a similar yield with higher quality compared to 100% ETc without NABF as well as increase WUE.

## 1. Introduction

*Solanum lycopersicum* L., i.e., the tomato, belongs to the Solanaceae family, which also includes eggplant, potato, pepper, nightshade, tobacco, and petunia. Throughout the world, tomatoes are a staple in most human diets [1]. They are a rich provenance of antioxidants (e.g., phenols, carotenes, lycopene, and vitamin C), minerals, carbohydrates, fats, proteins, and dietary fibers for most diets of humans. In addition, the importance and valuable phytochemical components of tomatoes make them highly favored by consumers worldwide [2,3,4]. Currently, worldwide fresh tomato fruit yield is approximately 0.152 billion tons from about 4400 thousand hectares most of which are located in China, India, USA, Turkey, and Egypt, those being the top five producing countries [5]. Tomatoes demand a lot of water [6], particularly in regions with a Mediterranean climate [7]. 

The biggest obstacle to global agricultural sustainability and food security is the shortage of water for irrigation purposes. Due to the global climatic changes and rapidly expanding population, there is an increasing demand for freshwater resources in the future [8]. Currently, significant freshwater shortage issues are raised due to improper irrigation water management [9]. There is insufficient rainfall in many tomato-growing regions, including Egypt, to meet the crop’s needs for water. Consequently, irrigation is required to prevent crop cultivation failure or minimization of productivity. However, it is vital to develop novel techniques for the management of water. The aim is to save water while at the same time maintaining appropriate production levels due to the rising water demand from other sectors and the anticipated future decline in water availability [10]. Undoubtedly, one of these methods is the use of deficit irrigation (DI), which purposefully maintains a certain level of water deficit and yield loss [11,12]. With this method, it is anticipated that any yield loss will not be as significant as the advantages of water conservation. With varying degrees of success [13,14], the effects of DI have been thoroughly examined in several crops, including tomatoes [15,16,17]. Along with irrigation management techniques, numerous studies have been conducted in recent years to examine ways to conserve water by utilizing the integral characteristics of some agrochemicals typically used for plant protection, such as fertilizers [18,19,20] and antioxidants [21,22,23,24], with the ultimate goal of enhancing the water use efficiency (WUE) of a crop. Plants’ ability to respond to stress is greatly influenced by their mineral and nutritional status. Nutrient management techniques can be utilized to produce a satisfactory crop yield under DI. Kumari et al. [25] reported that nutrients help plants to stimulate numerous plant mechanisms to relieve abiotic stress, including biosynthesis of antioxidant enzymes, growth, osmoprotectants (OPs), activation of stress-related genes, ROS suppression, protein functional protection, DNA repair, membrane stability, and improved photosynthetic capacity. Management of nutrients is an applicable technique for suppressing stress impacts and maximizing cropland productivity. Many researchers reported that application of N, P, K, Ca, Mg, Se, and/or I alleviate the damage of drought stress (DS) on tomato and other plants [18,26,27,28,29,30]. According to Waraich et al. [31], the addition of different nutrients (i.e., B, Cu, Si, Mg, Zn, Ca, K, P, and N) reduces the toxicity of ROS by increasing the content of antioxidants and the activity of enzymes (i.e., SOD, CAT, POD, APX, GR, etc.). These findings encourage enhanced root growth, which in turn increases water intake and improves DS tolerance in crops. Additionally, they noted that applying nutrients like K and Ca helps maintain high tissue water potential during DS conditions, improves tolerance of DS through osmotic adjustment, and indirectly attenuates the negative damage of DS by activating physiological, biochemical, and metabolic processes in plants. Plant antioxidant defense systems regulate the equilibrium between ROS formation and consumption when growth conditions are optimal [32]. However, under severe water stress, the antioxidant capability may not be adequate to reduce the injurious effects of oxidative damage. Therefore, to boost crop plants’ productivity with restricted water supply, it is vital to increase their resistance to drought stress. Exogenous applications of antioxidants efficiently attenuate the damage of DS in plants. More attention is being paid recently to safe and natural antioxidant substances such as glutathione (GSH), ascorbic acid (AsA), salicylic acid (SA), and citric acid (CA) which can eliminate free radicals, increasing plants’ resiliency to stress. Also, they added that the use of such antioxidants may enhance different parameters like photosynthetic efficiency, leaf tissue cell integrity, leaf macro- and micronutrients, OPs, and enzymatic antioxidant activities as well as soluble protein increasing DS tolerance in plants [10,33,34,35].

Given the pivotal role of antioxidants and nutrients in resisting environmental stresses we innovated a new component called “a novel nutrient- and antioxidant-based formulation (NABF)”; it is rich in different nutrients (I, Se, Mg, Ca, K, P, and N), vitamins, and antioxidants (i.e., citric acid, salicylic acid, ascorbic acid, glutathione, and EDTA). The current study used NABF as a foliar nourishment to suppress the damage of DS (80 or 60% of crop evapotranspiration; ETc) in tomato plants cultivated under soil salinity stress (SSS, EC = 10.2 dS m^−1^). This is the first report that investigated the impact of NABF in alleviating DS impacts on salinity-stressed tomato plants. Determining whether exogenously supplied NABF could reduce DS impacts in salt-stressed tomato plants was the goal of our investigation. According to our hypothesis, exogenous application of NABF would increase the tomato plants’ ability to withstand the impacts of the stressors under study by promoting plant development, altering plant water status by increasing OPs and antioxidant levels, and strengthening enzymatic and non-enzymatic antioxidant activities.

## 2. Results

### 2.1. Growth and Yield Traits, and Water Use Efficiency

Our results show tomato growth traits including plant leaf number (PLN), plant leaf area (PLA), and plant shoot dry weight (PSDW) in response to irrigation (ETc), a novel nutrient- and antioxidant-based formulation (NABF), and the ETc × NABF interaction (Table 1). Plants that experienced moderate or severe drought stress (DS) (ETc_80%_ and ETc_60%_, respectively) had lower PLN, PLA, and PSDW than those fully irrigated (ETc_100%_). However, NABF-treated plants had higher PLN, PLA, and PSDW than those obtained from untreated plants. These growth traits were reduced gradually with increasing DS (from ETc_80%_ to ETc_60%_) resulted in decreases of PLN by 21.3 and 43.8%, PLA by 21.5 and 25.6%, and PSDW by 25.6 and 40.1% as compared with those of ETc_100%_. Otherwise, NABF-treated tomato plants significantly increased PLN, PLA, PSDW, and WUE by 47.1%, 47.8%, 59.6%, and 120.3%, respectively, relative to untreated plants. The combined NABF × ETc_100%_ or NAB F × ETc_80%_ treatment increased PLN (by 33.9 or 7.3%), PLA (by 33.8 or 6.8%), and PSDW (by 56.6 or 12.3%), compared to ETc_100%_ without NABF. Moreover, no significant difference was found between plants treated with NABF × ETc_60%_ and plants receiving ETc_100%_ without NABF. As presented in Table 1, DS, NABF, and their interaction had distinct impacts on tomato yield traits. Drought stress (ETc_80%_ and ETc_60%_) decreased plant fruit weight (PFW) by 42.0 or 59.2%, plant fruit number (PFN) by 35.5 or 51.7%, fruit yield per ha (HFY) by 40.6 or 58.7%, and water use efficiency (WUE) by 27.2 or 22.5% compared to ETc_100%_. Fruit yield components of tomato plants responded to treatments and were raised with NABF application by 120.1, 96.0, and 129.0%, respectively, relative to the absence of NABF. Table 1 showed maximized values of PFW (9.3 kg), PFN (54.1), HFY (121.2 ton), and WUE (21.29 kg m^−3^), which were noted in the NABF × ETc_100%_ treatment, whereas the minimum values of PFW (0.81 kg), PFN (10.1), and HFY (10.2 ton) were observed under ETc_60%_ without NABF.

### 2.2. Tomato Fruit Quality Traits 

Table 2 shows the influences of DS, NABF, and their interaction on the fruit quality of the tomato such as vitamin C (Vit C), total soluble solids (TSS), titratable acidity (TA, % citric acid), lycopene, β-carotene (β-Car), firmness, and contents of selenium and iodine. Some of the fruit quality parameters were enhanced considerably under DS. The ETc_60%_ increased Vit C by 22.86%, TSS by 17.7%, TA by 52.4%, lycopene by 25.7%, and β-Car by 41% relative to the ETc_100%_. Firmness and the contents of Se and iodine, on the other side, were decreased by 9.2%, 22.7%, and 24.4%, respectively, in the ETc_60%_-exposed plants relative to those in fully irrigated plants. Concerning the effects of NABF treatments, all of the above variables were markedly augmented by the application of NABF compared to untreated plants. The increases generated by NABF were 10.53% for Vit C, 8.77% for TSS, 18.37% for TA, 8.85% for lycopene, 13.32% for β-Car, 10.14% for firmness, 30.16% for Se, and 32.12% for iodine compared to the controls. The NABF × ETc_60%_ treatment gave the highest values for Vit C, TSS, TA, and lycopene, while the ETc_100%_ treatment without NABF showed the lowest values. Otherwise, the NABF × ETc_100%_ treatment conferred the highest values of firmness, Se, and iodine while the lowest values were noted under ETc_60%_ with the absence of NABF.

### 2.3. Leaf Photosynthetic Pigments

Responses of photosynthetic efficiency (i.e., relative content of chlorophyll (SPAD value), chlorophyll and carotenoid contents, chlorophyll fluorescence (Fv/Fm and PI), and photochemical activity (PhAc)) of tomato plants to DS with or without the novel nutrient- and antioxidant-based formulation (NABF) treatment are in Table 3. The components of photosynthetic efficiency were noticeably impacted by watering levels with or without NABF treatment. Moderate or severe DS (ETc_80%_ or ETc_60%_, respectively) significantly reduced SPAD by 9.1 or 26.2%, chlorophyll content by 7.6 or 20.8%, carotenoid content by 9.6 or 27.7%, Fv/Fm by 6.0 or 12.0%, PI by 8.9 or 20.3%, and PhAc by 8.6 or 22.9%, respectively, relative to ETc_100%_ treatment, while tomato plants treated by NABF positively increased SPAD by 31.3%, chlorophyll content by 23.3%, carotenoid content by 34.0%, Fv/Fm by 14.1%, PI by 22.6%, and PhAc by 27.3%, respectively, compared to untreated plants. SPAD, chlorophyll and carotenoid contents, Fv/Fm, PI, and PhAc were notably impacted by the interaction between irrigation and NABF treatments. Maximum values were recorded with ETc_100%_ × NABF treatment, while minimum values were noted with ETc_60%_ without NABF treatment. The statistical analysis for leaf photosynthetic efficiency traits showed that the differences between ETc_100%_ and ETc_60%_ × NABF treatments were not significant.

### 2.4. Leaf Macro Mineral Content

As average for both growing seasons, irrigation levels, NABF treatment, and their interaction had considerable (*p* ≤ 0.05) impacts on the leaf macro-mineral contents of tomato plants grown in saline soil (Table 4). When tomato plants were exposed to moderate (ETc_80%_) or severe deficit irrigation (ETc_60%_) level, N, P, K^+^, Ca^2+^, Mg^2+^, and S reduced by 21.3 or 33.1%, 11.1 or 24.0%, 12.0 or 30.0%, 13.4 or 30.8%, 11.2 or 26.9%, and 18.4 or 36.2%, respectively, compared to fully irrigated plants. Compared to the untreated plants, nutrient uptake of stressed tomato plants treated with NABF significantly (*p* ≤ 0.05) increased by 50.6% for N, 31.1% for P, 45.5 for K^+^, 46.7% for Ca^2+^, 34.9 for Mg^2+^, and 55.9% for S. The combined NABF × ETc_100%_ application gave the highest values of all macronutrients, while the treatment ETc_60%_ without NABF gave the lowest values. Otherwise, the two treatments NABF × ETc_60%_ and ETc_100%_ without NABF showed negligible differences for all aforementioned traits.

### 2.5. Leaf Micro Mineral Contents

Drought, NABF application, and their interactions (ETc × NABF) significantly (*p* ≤ 0.05) affected leaf micronutrient (Fe, Mn, Zn, Cu, Se, and iodine) levels of tomato plants that experienced DS (Table 5). Compared to ETc_100%_, ETc_80%_ or ETc_60%_ significantly (*p* ≤ 0.05) reduced Fe by 11.6 or 26.4%, Mn by 9.5 or 20.9%, Zn by 11.0 or 25.0%, Cu by 9.8 or 20.3%, Se by 13.2 or 25.2%, and iodine by 12.4 or 25.7%, respectively. It is worth noting that salinity-stressed plants treated with NABF statistically improved (*p* ≤ 0.05) micronutrients by 34.9%, 25.7%, 32.4%, 26.5%, 32.1%, and 33.9%, respectively, relative to un-supplemented salinity-stressed plants. Concerning the ETc × NABF interaction, the best results for leaf micro mineral contents were noted with ETc_100%_ × NABF interactions, which significantly (*p* ≤ 0.05) improved Fe by 20.2%, Mn by 14.7%, Zn by 25.3%, Cu by 21.7%, Se by 24.7%, and iodine by 25.4% compared to ETc_100%_ without NABF (Table 5). Additionally, no significant differences were observed for leaf micro mineral contents between ETc_100%_ without NABF and ETc_60%_ × NABF.

### 2.6. Integrity of Leaves

Tomato plants that experienced DS showed a marked decrease in leafy relative water content (RWC) and membrane integrity index (MSI) values (Figure 1). As compared to fully irrigated plants (ETc_100%_), DS (at ETc_80%_ or ETc_60%_) significantly (*p* ≤ 0.05) reduced RWC by 6.7 or 12.9% and MSI by 7.3 or 14.9%, respectively. Nevertheless, NABF treatment lessened DS-stimulated damage to the leaf. When tomato plants were supplemented by NABF, an increase in RWC and MSI by 14.2 and 17.5% were noted, respectively. The integrative NABF × ETc_100%_ stimulated ameliorating impacts on DS-damaged leaves, increasing the RWC and MSI by 12.4% and 16.1%, respectively, relative to the fully irrigated plants without elicitors (NABF).

### 2.7. Ionic Leakage, Lipid Peroxidation, and Oxidative Stress Markers

The EL and oxidative stress indicators (i.e., EL, MDA, H_2_O_2_, and O_2_^•−^ level), were determined to assess the leaf oxidant levels of the tomato plant against DS. In this sense, the obtained results (Figure 2) reveal that the DS caused by ETc_60%_ or ETc_80%_ treatments noticeably (*p* ≤ 0.05) increased EL by 32.7 or 56.1%, MDA by 23.6 or 57.3%, H_2_O_2_ by 20.3 or 43.0%, and O_2_^•−^ by 27.2 or 62.4%, respectively, compared to plants irrigated at ETc_100%_. Nevertheless, under tested watering levels, NABF treatment decreased EL by 34.5%, MDA level by 37.6%, H_2_O_2_ by 30.2, and O_2_^•−^ by 38.7%. Under full irrigation (ETc_100%_), the decreases were 28.4% and 22.6%, and 23.5%, and 16.2%, respectively, relative to the respective control. The highest decreases for the above-mentioned traits were noted when plants were exposed to severe irrigation (ETc_60%_) × NABF treatment. Furthermore, these decreases were 41.6% for EL, 45.5% for MDA, 35.1% for H_2_O_2_, and 50.7% for O_2_^•−^ compared to the corresponding controls.

### 2.8. Osmoprotectants (OPs) and Antioxidant Compounds

OPs (i.e., S. sugar content and proline) and antioxidants (i.e., AsA, ascorbate; GSH, glutathione; and αToc, α-Tocopherol) were determined to assess the antioxidant defense system of the tomato plant against DS. In this regard, the data in Table 6 reveal that the DS caused by ETc_80%_ or ETc_60%_ treatment considerably (*p* ≤ 0.05) increased S. sugar content by 17.4 or 38.89%, proline by 42.3 or 103.9%, AsA by 14.8 or 39.1%, GSH by 18.7 or 119.3%, and αToc by 12.9 or 31.9%, respectively, compared to ETc_100%_ treatment. Furthermore, salinity-stressed plants treated with NABF significantly enhanced S. sugar content by 17.9%, proline by 18.9%, AsA by 17.3%, GSH by 14.5%, and αToc by 14.1% compared to unamended soil. The ETc × NABF interaction had a significant impact on OPs and antioxidant compounds of tomato plants cultivated in saline soil (Table 6). The best contents of S. sugars (35.4 mg g^−1^ DW), free proline (0.29 mg g^−1^ DW), AsA (2.74 µM g^−1^ FW), GSH (1.98 µM g^−1^ FW), and αToc (3.42 µM g^−1^ DW) in tomato leaves were noted under ETc_60%_ × NABF vs. the lowest contents (21.6 and 0.12 mg g^−1^ DW, 1.70 and 0.80 µM g^−1^ FW, and 2.28 µM g^−1^ DW, respectively), which were noted under ETc_100%_ without NABF treatment.

### 2.9. Enzymatic Antioxidant Activities and Soluble Protein Content (SPC)

Table 7 show the responses of the tomato plant’s enzymatic antioxidants and SPC to DS (ETc_80%_ or ETc_60%_) strategy, NABF treatment, and their interaction. Drought stress × ETc_80%_ or ETc_60%_ increased (*p* ≤ 0.05) the activity of SOD by 14.9 or 53.9%, CAT by 33.1 or 98.1%, APX by 15.7 or 64.2%, and GR by 19.2 or 53.7%, while it decreased SPC by 9.7 or 21.2%, respectively, compared to ETc_100%_. Moreover, an increment was noticed in the activation of SOD, CAT, APX, GR, and SPC when tomato plants were supplemented with NABF. The nourishment of plants with NABF led to an increase in SOD, CAT, APX, GR, and SPC activities of leaves by 16.2, 20.8, 17.9, 21.4, and 25.6%, respectively, as compared with plants untreated by NABF. The ETc × NABF interaction was significant (*p* ≤ 0.05) for the activity of SOD, CAT, APX, GR, and SPC of tomato plants under studied stressful conditions (Table 7). Under ETc_60%_ × NABF interaction, the increments in the above traits were 78.3, 132.8, 91.4, 82.5, and 31.4%, respectively, relative to those noted under ETc_100%_ without NABF treatment.

### 2.10. Levels of Transcription of Antioxidant Enzyme-Encoding Genes of Tomato Plants

Data in Figure 3 illustrate the effects of the irrigation levels, NABF, and their interaction on the levels of transcription of antioxidant enzyme-encoding genes of tomato plants. Plants irrigated at ETc_100%_ had lower values of gene transcription levels of *SOD*, *CAT*, *APX*, and *GR* in tomato plants than those stressed with ETc_80%_ or ETc_60%_. These traits were increased significantly with increasing DS (form ETc_80%_ to ETc_60%_), resulting in an increment in the relative expression of *SOD* by 30.4 or 95.6%, *CAT* by 39.4 or 99.1%, *APX* by 38.2 or 115.5%, and *GR* by 24.0 or 83.3% compared to ETc_100%_. Plants supplemented by NABF favorably increased their relative expression of *SOD* by 32.5%, *CAT* by 26.4%, *APX* by 40.2%, and *GR* by 29.6%. The NABF × ETc_60%_ treatment conferred the best gene transcription levels in tomato plants, while the ETc_100%_ without NABF treatment gave the lowest ones. Further, the NABF × ETc_60%_ treatment conferred better values than ETc_100%_ without NABF.

### 2.11. Phytohormone Levels

The analyses of hormonal contents (IAA, GA_3_, CKs, and ABA) of tomato plants grown in saline soil displayed differences between the DS, NABF treatment, and their interactions (Figure 4). Drought-stressed plants exhibited lower values of IAA, GA_3_, and CKs, and higher ABA values than unstressed plants. In this respect, the results in Figure 4 show that the DS caused by ETc_80%_ or ETc_60%_ treatments considerably (*p* ≤ 0.05) reduced IAA by 11.1 or 28.7%, GA_3_ by 10.0 or 33.2%, and CKs by 17.4 or 33.2%, while it increased ABA by 31.2 or 69.2%, respectively, compared to ETc_100%_. As for NABF, treated tomato plants showed higher IAA, GA_3_, and CKs, and lower ABA contents than untreated plants. In this respect, the obtained results (Figure 4) revealed that NABF-supplemented plants revealed marked increases (*p* ≤ 0.05) at 42.4% for IAA, 26.4% for GA_3_, and 48.6% for CKs, while the ABA level was decreased by 41.8% compared to un-supplemented plants. The NABF × ETc_100%_ recorded the maximum values of IAA (23.2 µg kg^−1^ FW), GA_3_ (42.8 µg kg^−1^ FW), and CKs (33.2 µM g^−1^ FW), while the highest value of ABA (9.9 µg kg^−1^ FW) was recorded under ETc_60%_ without NABF application. On the other hand, the minimum values of IAA (11.1 µg kg^−1^ FW), GA_3_ (26.1 µg kg^−1^ FW), and CKs (33.2 µM g^−1^ FW) were noted under ETc_60%_ × NABF, while the lowest value of ABA (3.66 µg kg^−1^ FW) was observed under ETc_100%_ × NABF.

## 3. Discussion

As anticipated, lowering the water content in saline soils (10.2 dS m^−1^; Table 8) noticeably lowered the growth of tomato plants. Applying a novel nutrient- and antioxidant-based formulation (NABF) as a foliar spray induces apparent positive outcomes in growth and yield traits (Table 1), fruit quality (Table 2), photosynthetic efficiency (Table 3), leaf macro- and micronutrient contents (Table 4 and Table 5), leaf tissue cell integrity (Figure 1), leaf oxidant levels (Figure 2), osmoprotectants (OPs) and antioxidant compounds (Table 6), enzyme activities (Table 7), transcription levels of enzymatic genes (Figure 3), and hormonal contents (Figure 4) of tomato plants that experienced both drought stress (DS) and soil salinity stress (SSS) (Table 8). This report reports the stimulation of NABF, as a multiple growth stimulator of *Solanum lycopersicum* plants to tolerate both DS and SSS through its foliar nourishment. After leaf nourishment, the NABF ingredients (Table 9) can easily penetrate the stomata of sprayed leaves and translocate to bioactive tissues to provide them with the potential to attenuate the stress influences. The findings of this report propose that NABF contributed to serviceable roles during tomato plant growth by promoting metabolic processes due to the high NABF content of essential nutrients and antioxidants. These nutrients and antioxidants are reported to promote division and elongation of plant cells and dry matter accumulation [36,37], which enable tomato plants to relieve the damage of both DS and SSS, contributing markedly to improving growth and yield traits, and fruit quality under DS and SSS conditions (Table 1).

As previously studied individually, nutrients and antioxidants, as growth multiple stimulators, confer specific merits, including enhancements in growth traits, productivity, yield quality traits, and better modifications of plant physio-biochemical processes [38,39]. Maximizing WUE is an important strategy for mitigation the global water shortage, which can be achieved by increasing crop yields per consumed water unit. In the irrigated agricultural regions, increasing WUE is more crucial for growers than increasing yield potential per unit area [10]. Tomato plants treated with NABF increased WUE by 120% relative to non-treated plants.

Nutrients clearly influence a plant’s ability to cope with the effects of environmental stress through signaling pathways, which influence plant adaptive responses to stress and/or upregulate stress-related genes to increase stress tolerance [40]. In addition, vitamins activate endogenous morphogenesis and upregulate physiological correlations among different plant organs [41]. Different antioxidants (i.e., glutathione, salicylic acid, EDTA, and ascorbic acid) had key functions in plant response to abiotic stressors and regulation of plant development and growth [42,43,44,45].

Drought stress notably decreases photosynthetic efficiency traits in terms of relative content of chlorophyll (SPAD), chlorophyll, carotenoids, photosystem II quantum efficiency (Fv/Fm), performance index (PI), and photochemical activity (PhAc). During DS, the reduction of chlorophyll content accelerates the decrease in the photochemical activity of chloroplasts, which is responsible for the reduction in the efficiency of photosynthesis. The impacts of DS on total photosynthetic efficiency traits are consistent with some reports [46,47]. As revealed by Nankishore and Farrell [48], DS notably diminished chlorophyll in tomato leaves. Nutrient (i.e., N, P, K, Mg, Ca, and Se) provision has significant positive effects on SPAD, chlorophyll, carotenoids, Fv/Fm, PI, and photochemical activity under full watering or DS, which were previously reported with rice, tomato, onion, and eggplant [49,50,51,52]. Drought stress resulted in decreased relative chlorophyll content and photosynthetic capacity (Fv/Fm and PI) in PSII, which could be due to decreased uptake of essential nutrients for chlorophyll biosynthesis [52,53], concomitant with inhibition and degradation of protein synthesis and thylakoid membranes, respectively owing to oxidative stress under DS [54]. These indicate inhibition of electron transport chains and the PSII light-harvesting complex in bean plants that experienced DS. However, the antioxidants in the complex (NABF) used in the current study (e.g., AsA SA, GSH, and CA) probably mediated the restoration of the relative content of chlorophyll and capacity of photosynthesis of tomato plants, and revealed that NABF elevated SPAD reading, chlorophyll, carotenoids, Fv/Fm, PI, and PhAc values (Table 3). These results were linked to enhancement of water status (RWC) and membrane stabilization (MSI) in plants (Figure 1) by NABF to restore damaged chloroplasts and raise chlorophyll contents [21,55]. These results support those of [34,56,57]. They proved that treating plants with AsA, GSH, SA, and CA caused the increased SPAD reading, chlorophyll, carotenoids, Fv/Fm, and PI of barley and beans under abiotic stress. Our findings showed that water shortage stress had a negative impact on tomato plants as indicated by decreased levels of macronutrients (i.e., N, P, K, Ca, Mg, and S) and micronutrients (i.e., Fe, Mn, Zn, Cu, Se, and I) in leaves (Table 8 and Table 9); consequently, this may have caused the ionic imbalance and nutrient deficiency [58,59]. They demonstrated that plants cultivated in water deficits exhibit restricted nutrient uptake as a general phenomenon. According to [60], nitrogen absorption by roots diminishes when water availability is constrained because a drop in soil water potential lowers the pace at which nutrients diffuse from the soil matrix to the root surface. Plants experiencing DS have decreased rates of transpiration and active transport as a result of decreased energy expenditure and altered permeability of membranes [61]. Nonetheless, exogenous application of minerals and antioxidants could improve these ion disorders by elevating macro- and micronutrient uptakes as another mechanism of drought tolerance by which tomato plants endeavor to keep ion homeostasis and turgor of leaf cells [62,63]. These desirable effects on nutrient accumulation in tomatoes can be attributed to the different roles of minerals or antioxidants in improving the integrity of cell membranes. This leads to improved ion selectivity and transport. In addition, the accumulation of K^+^ ions and osmo-regulatory compounds contributes to an increased influx of water and nutrients [64]. Our results are in agreement with those of [27,28,29,65,66,67,68,69,70,71,72]. They reported that absorption of macro- and micronutrients (I, Se, Mg, Ca, K, P, N, etc.) is elevated by applying these nutrients alone or together under normal or DS conditions. Na- and Cl-dominated salinity not only decreases the availability of calcium and phosphorus (Ca and K) but also hinders their transport and mobility to the plant’s growth regions, which has an impact on the quality of both the plant’s vegetative and reproductive organs. In addition, salt stress decreases the amounts of magnesium, manganese, iron, zinc, and copper in a variety of crops [19]. Macro- and micronutrient content in tomato plants, in this study, was found to decrease due to the high salinity of the tested soil (10.2 dS m^−1^). However, these values were affirmatively impacted by the exogenous AsA, GSH, salicylic acid (SA), and citric acid (CA) which were included in NABF applied to the tomato plants. In addition, under water stress, the plant media supplemented with NABF showed a significant increase in macro and micro mineral levels when it was compared with untreated plants (Table 8 and Table 9). These findings supported those obtained previously in broccoli and strawberry plants [33,67]. The impact of antioxidants on nutrient absorption may be attributed to their capability in enzyme activity regulation in secondary metabolic pathways and phenolic biosynthesis [68], which includes cell wall lignification and attenuates the osmo-regulatory stress [69,70]. In our study, dropping the level of irrigation from 100% to 60% ETc limited tomato growth and production (Table 1), weakened the efficiency of photosynthetic machinery (Table 3), and damaged leaf integrity, i.e., RWC and MSI (Figure 1). Consequently, MDA signalizing lipid oxidation was elevated (Figure 2) due to the over-increase of oxidative stress marker levels (H_2_O_2_ and O_2_^•−^). This was shared with elevated OP compounds (Table 6), which tolerate oxidative damage under DS [71]. The negative impacts worsened by DS could be attributed to loss of cell turgor under osmotic stress and/or increased ROS production under DS [72,73].

Drought stress induces oxidative damage; this was evidenced by the increase in EL, MDA, O_2_^•−^ level, and ROS accumulation (i.e., H_2_O_2_ content) in tomato leaves. This could be the reason for the membrane damage and lipid peroxidation in plant cells [74,75], thereby reducing tomato leaf water relations.

In this study, the machinery of plant defenses, including synthesis of soluble protein, OPs, and antioxidant compounds (S. sugars, proline, αToc, GSH, and AsA; Table 6), and activation of antioxidative enzymes (SOD, CAT, APX, and GR) was elevated in plants treated with NABF. This desirable status safeguarded tomato plants against the damage of DS through osmotic modification and ROS elimination [76,77]. The high level of ROS under DS caused a decrease in the photosynthesis rate (Table 7) [78].

Typically, the plant accumulates various OPs in the cytosol of cells to increase the osmotic potential to withstand DS, improving or maintaining cell turgidity for continuing processes related to plant growth. Further, these OPs contribute to detoxifying ROS, stabilizing membranes, and protecting macromolecules [79]. Our findings were that drought-stressed plants accumulated more soluble sugars and proline than normal plants. However, under DS, soluble protein content was lower than that in normal plants. This may be due to a shared elevated function of protease enzymes, proteolysis, or lowered synthesis of proteins in plants under DS. Additionally, NABF further increased the contents of soluble sugars, proline, and soluble protein in drought-stressed plants compared with those in NABF-free applications [80]. Kim et al. [27] reported that an elevated soluble sugar concentration with nutrient application under water scarcity is caused by the decrease in normal soluble sugar transport, use, and distribution during DS, or by the hydrolysis of starch [19]. Also, a higher proline accumulation is noticed in potato plants nourished by nano-Mg. However, an affirmative role has been found for N application in reducing N-compound assimilation [81]. Our outcomes signalized that tomato cells can keep their osmotic potential under DS; however, NABF maximized cell osmotic potential and RWC, which maximized plant DS tolerance.

Compared to enzyme activity, gene expressions in response to abiotic stresses give a more accurate measure of gene activation [82]. Our findings are similar to those reported by Yu et al. [83] in lupin plants; they stated that antioxidants upregulate these genes’ expression patterns differently in response to water stress. Therefore, under DS, NABF treatment promotes the tomato plant’s antioxidant defense mechanisms, which were worsened enough to eliminate ROS and minimize the level of MDA for the NABF-free drought-stressed plants. Intriguingly, NABF noticeably minimized the production of MDA and O_2_^•−^ due to noticeably increased SOD and POD activities. Concisely, our outcomes signalize that the tomato plant’s antioxidant defense mechanisms have a lower impact when responding to DS-stimulated damage; however, via NABF, these mechanisms were considerably strengthened to overcome the undesirable impacts of DS.

It has been well revealed that hormonal contents (IAA, GA_3_, CKs (cytokinins), and ABA) play an affirmative role in different processes related to plant morpho-physio-biochemistry and molecular biology to alleviate DS [84], which was noticeably elevated by NABF treatment, while ABA content was decreased. In this concern, NABF supported IAA, CK, and GA_3_ contents in drought-stressed tomato plants. This may be due to the accumulated macro- and micronutrients necessary for protoplasm formation and phytohormone biosynthesis [85,86]. The reports [80,87,88,89] point out that plants treated with fertilizers (i.e., urea and potassium) or antioxidants (i.e., glutathione or salicylic acid) led to higher levels of CKs, IAA, and GA_3_ while reducing ABA. Our outcomes signalized that NABF improved the tomato plant self-protection mechanisms under DS by exerting an impact on the key enzyme activities and certain hormone levels. Finally, the detrimental consequences of environmental threats might be greater than stressed plants can withstand. In this situation, the drought-stressed plant’s defense systems are insufficient to meet the requirements for self-defense, so exogenous contributory substances, such as fertilizers and/or antioxidants, and other advantageous techniques, must be applied to maximize the effectiveness of the plant’s defenses. As a result, plants can function effectively under challenging environmental conditions [51].

## 4. Materials and Methods

### 4.1. Trial Location and Soil Analysis

A loam soil piece of 760 m^2^ in a private farm (Sedmant El-Gabal; 29.8870159 N, 31.0640144 E), Fayoum, Egypt, was assigned for two attempted field studies in two consecutive early summers of 2021 and 2022. Table 10 shows the average data of the experimental area weather.

Each season, soil samples were taken from the top 0–30 cm of the soil before transplantation. The hydrometer method was used to determine the distribution of particle size [90]. According to Page et al. [91], the soil pH was measured in saturated soil–water paste using a Bekman pH meter (model Elico, LI120-UK). Using a CM25 conductivity meter (model 3200, YSI, Inc., Yellow Springs, OH, USA), the ECe values were calculated in saturated soil–water paste extract and quantified as dS m^−1^, as stated by Page et al. [91]. According to Page et al. [91], the Collin Calcimeter method was used to calculate total CaCO_3_. Wet combustion was used to calculate the OM content of the soil in accordance with Walkly and Black’s approach [91]. Exchangeable cations were extracted with ammonium acetate (NH_4_OAc; 1 N). Na^+^ and K^+^ were determined using flame photometry (a Perkin-Elmer Model 52-A Flame Photometer), while Ca and Mg were determined using the EDTA titration method. The method of Livens [92] was used to calculate the amount of available N in the soil. Available P was extracted by 0.5 N sodium bicarbonate solution (NaHCO_3_) at pH 8.5 and the extract was determined using a PerkinElmer Model 3300 Atomic absorption Spectrophotometer at wavelength 660 [93]. For 30 min, available K was extracted from soil sample by shaking with 1 N ammonium acetate solution and determined by flame photometry (Perkin-Elmer Model 52-A Flame Photometer). Available micronutrients (Fe, Mn, Zn, and Cu) were extracted by diethylene triamine penta acetic acid method (DTPA). Through a Whatman No. 42 filter paper, the extract was filtered. Then, micronutrients in the extract were measured using atomic absorption spectrophotometer (Perkin-Elmer, Model 3300). Table 8 shows the analysis data, where the EC of the soil paste extract was 10.2 dS m^−1^, indicating that it is salt-affected soil [94], and the texture was loam with an organic matter content of 0.98%. The aridity index pointed out that the farm is located in arid region [95].

### 4.2. Transplanting, Experimental Treatments, and Layout

Forty-day-old *Solanum lycopersicum* L. transplants (cultivar 320^®^) were secured by the Nurseries of Agriculture Ministry, Cairo, Egypt. The seeds were produced by Seminis company, St. Louis, MO, USA. After the transplants were sorted to ensure their authenticity and symmetry, they were transplanted on February 20 for the two seasons (2022 and 2023) in mounds 50 cm apart, each mound containing one transplant. The experiments were designated as a split-plot arrangement for six treatments with 3 replicates, and each replicate was represented by a row 15 m long and 1.5 m wide, which contained 30 plants. Two factors were assigned; irrigation regimes (100, 80, and 60% of the crop evapotranspiration; ETc) were the first factor intended for main plots, while the novel nutrient- and antioxidant-based formulation (NABF) was the second factor intended for sub-plots. The NABF was applied as foliar nourishment at 0 (control, foliar spray with distilled water) and 2.5 g L^−1^. This concentration (2.5 g L^−1^) of the NABF was selected based on a preliminary study, in which five times foliar nourishment with 2.5 g NABF L^−1^ gave the best tomato growth (plant leaf number, plant leaf area, dry weight of plant shoot) results among several concentrations used. Additionally, this formulation was selected from several formulations due to its best results. The NABF composition is presented in Table 9. 

The irrigation regimes (full irrigation; 100% of ETc and drought stress (DS); 80% of ETc (moderate DS); and 60% of ETc (severe DS)) were separated by 2 m of the non-irrigated area to avoid interference. Until the transplant roots were well repaired and fixed in the rhizosphere (two weeks after transplanting; WAT), tomato transplants of all treatments were provided with 100% ETc. With the start of the irrigation treatments (2 WAT), foliar feeding with NABF was applied once every 10 days five times (the vegetative and flowering stages). The NABF solution was sprayed until runoff using a 20-L Dorsal Sprayer. The share of each plant of the NABF was 0.45 g of the 5 leaf sprays. A surfactant of 1 mL Tween-20 per L of spray solution was provided to ensure optimal entry into leaf tissues. According to the recommendations of the Egyptian Agricultural Research Center, all agricultural practices, including weed and disease control were applied.

### 4.3. Fertilization Program 

The tomato fertilization program detailed in Rady et al. [96] was used with some modifications as follows: For a month starting a week after transplantation, 1.5 g of NPK fertilizer (Super feid 20/20/20, Technogreen Co., Nakazakinishi, Japan) per L was added day after day. Three grams humic acid (Humutech 45%, Technogreen Co.) + 3 g Ca(CO_3_)_2_ (15.5/0/0 + 26 Cao, Evergrow Co., Cairo, Egypt) per L were added to the soil once a week. Once a week also, 2 cm amino acids (Aminoplus TG 22.5%, Technogreen Co.) per L + 2 g micronutrient mixture (Fedex, Pharmaceutica Co., Mississauga, ON, Canada) per L were sprayed. Additionally, for another month starting 6 weeks after transplantation, the above fertilizer rates were increased. NPK compound was increased to 5 g L^−1^, and humic acid, Ca(CO_3_)_2_, amino acids, and micronutrients were increased to 5 g L^−1^ each for the same addition times. Beginning from the 10th week after transplantation, K was increased to be added once a day.

### 4.4. Irrigation Water Applied (IWA)

Reference evapotranspiration (*ETo*) was positioned using the class A pan (*Epan* (mm per day)) data, with adjoining plots modified for a fitting pan coefficient (*K_pan_*) and crop coefficient (*K_c_*) [97]. The ETc (in mm day^−1^) was assessed by applying Allen’s equation [97]: *ETc* = *E_pan_* × *K_pan_* × *K_c_*

*IWA* (m^3^) was calculated using the following equation:$$IWA=\frac{A\times ET_{c}\times I_{i}\times K_{r}}{E_{a}\times 1000\times (1-LR)}$$
where *A* is area of the plot (m^2^), *ET_c_* is crop water needs (mm day^−1^), *I_i_* is irrigation intervals (day), *K_r_* is covering factor, *E_a_* is application efficiency (%), and *LR* is leaching needs.

The total *IWA* during the 2021 and 2022 seasons was 5684 and 5702, 4547 and 4562, and 3411 and 3421 m^3^ ha^−1^ for 100, 80, and 60% *ETc*, respectively. HH2 digital hygrometer sensors (Cambridge, UK) were used for soil water content at 2-day intervals at a 0–30 cm depth. Table 11 shows the chemical analysis of irrigation water. WUE was computed according to Fernández et al. [98] using the formula given below:$$WUE=\frac{Fruit\;yield({kg\;ha}^{-1})}{Water\;applied(m^{3}{\;ha}^{-1})}$$

### 4.5. Plant Sampling

Twelve weeks after transplantation (WAT), plant samples were taken. Randomly, 9 plants were taken from each treatment (3 plants from each row/replicate) for growth traits, photosynthetic efficiency parameters, and nutrient contents. In the same way, another 9 plants were taken and assigned to enzyme assays, enzyme gene expression, and hormonal analysis. A third set of 9 plants was taken randomly from each treatment and assigned to further analyses (leaf integrity parameters, oxidative stress markers, osmoprotectants (OPs), and non-enzymatic antioxidants). For each plant, after counting leaves, leaf area (cm^2^) was evaluated utilizing a held-hand Planix 7 planimeter (Tamaya Technics Inc., Tokyo, Japan). Shoots were oven-dried at 70 ± 2 °C for 48 h and the shoot dry weight was recorded.

### 4.6. Fruit Yield Characteristics, and Fruit Quality Traits

Beginning at 12 WAT, tomato fruits were collected five times, every seven days, from the remaining plants in each treatment. The fruits were used to evaluate the fruit yield components and quality of fruits. In the marketable tomato fruit stage, after counting plant fruits, fruit weight and yield (kg plant^−1^ and ton ha^−1^) were measured.

The homogeneous tomato fruits taken from the first harvest were used to assess fruit quality traits. Fruit content of lycopene and ascorbate (mg 100 g^−1^ fruit FW) were evaluated by applying the procedures of [99,100]. Fruit total soluble solids (TSS, °Brix) were evaluated utilizing a refractometer (Digital, PR-100, Atago Co. Ltd., Tokyo, Japan) [101]. Fruit content of selenium (Se; mg kg^−1^ FW) was evaluated applying the methods of [101,102]. Fruit content of iodine (I; mg kg^−1^ FW) was evaluated applying a standard method. Samples of dried tomato fruits were prepared for incubation with tetra methyl ammonium hydroxide (TMAH, 25%) and iodine assay was performed (prEN 15111: R2-P5-F01 2006 (E); Determination of iodine by ICP-MS (inductively coupled plasma mass spectrometry), Polish Committee of Standardization). According to the method of [103], tomato fruit firmness (kg cm^−1^) was measured utilizing a TA-XT2i Texture Analyser (Stable Micro Systems, Surrey, UK). Then, fruit juice was used to evaluate titratable acidity (as % citric acid) using a HI-422 digital pH meter (Hanna Instruments Inc., Woonsocket, RI, USA) with NaOH (0.1 N) titration. β-Carotene (mg 100 g^−1^ fruit FW) was evaluated by applying the procedures of [104]. A sample of 100 mg was mixed with 20 mL solution containing hexane and acetone at a ratio of 3:2. The absorbance was read for the supernatant at 663, 645, 505, and 453 nm, and the following equation was applied:β-Carotene = (0.216 × Abs_(663)_) − (1.22 × Abs_(645)_) − (0.304 × Abs_(505)_) + (0.452 × Abs_(453)_)

### 4.7. Assessment of Efficiency of Photosynthetic Machinery

The greenness of tomato leaf (SPAD) was measured utilizing a SPAD-502 Chlorophyll Meter (Minolta Sensing, Inc., Osaka, Japan). The procedures in [105] were followed to estimate the contents of total chlorophylls (TChls) and carotenoids (TCars) in mg g^−1^ fresh leaf tissue. TChls and TCars were extracted by homogenization of 100 mg sample in acetone solution (10 mL, 80%). The homogenates were centrifuged at 3000 rpm for 20 min. Overnight, the samples were stored, then the supernatant absorbance was taken at 480, 645, and 663 nm. Fresh leaf was utilized to estimate photochemical activity applying the ferricyanide technique [106]. The fluorescence of Chl ‘a’ was evaluated using a handy PEA Chl Fluorometer (Hansatech Instruments Ltd., Kings Lynn, UK). Fv/Fm (PSII maximum quantum yield) was evaluated using the equation (F_v_/F_m_ = (F_m_ − F_0_)/F_m_), where Fv is the variable fluorescence, Fm is the maximum fluorescence, and F0 is the basal fluorescence [107]. Performance index (PI) of photosynthesis was computed [108]. 

### 4.8. Nutrient Content Assessments

Nitric and perchloric acids were mixed at 3:1 (*v*/*v*) for use in the digestion of dried leaf samples. The digested solution was used to estimate macro- and micronutrient contents (mg g^−1^ DW). The procedures described in [101] were utilized to evaluate N content using micro-Kjeldahl apparatus (Ningbo Medical Instruments Co., Ningbo, China). The [93] procedures were utilized to evaluate P content depending on the reduction rate of H_3_PMo_12_O_40_ in H_2_SO_4_ by molybdenum to eliminate arsenic. K^+^ content was determined according to [109] using a flame photometer (Perkin-Elmer Model 52-A, Glenbrook, Stamford, CT, USA). The methods in [110] were utilized to evaluate Ca, Mg, and S (mg g^−1^ DW), as well as Zn, Mn, Fe, and Cu contents (mg kg^−1^ DW), against NIST (USA) standard reference samples using atomic absorption spectroscopy. In addition, Se and I in tomato leaves (mg kg^−1^ DW) were determined as described in Section 2.5 for tomato fruits.

### 4.9. Assessment of Leaf Tissue Integrity and Oxidative Stress Markers 

Relative water content (RWC), membrane stability index (MSI), and EL (electrolyte leakage) in leaf tissues were measured in line with [111,112,113]. The RWC determination was based on the weights of leaf blade fresh, dry, and turgid masses. MSI was calculated from the leaf tissue solution’s electrical conductivity (EC) under warm (40 °C) (EC1) and boiling (100 °C) (EC2) conditions. The leaf tissue solution EC under normal (EC1) room temperature (laboratory), warm (45–55 °C) (EC2), and boiling (100 °C) (EC3) conditions was used for EL measurements. The following formulas were applied: $$RWC\;\left(\%\right)=\left[\frac{\left(fresh\;mass-dry\;mass\right)}{\left(turgid\;mass-dry\;mass\right)}\right]\times 100$$
$$MSI\;\left(\%\right)=\left[1-(\frac{EC1}{EC2})\right]\times 100$$
$$EL\;\left(\%\right)=\left[\frac{\left(EC2-EC1\right)}{EC3}\right]\times 100$$

Lipid peroxidation (evaluated as MDA), superoxide (O_2_^•−^), and hydrogen peroxide (H_2_O_2_) levels (µM g^−1^ FW) were calculated as described in [114,115,116].

### 4.10. Assessments of Osmoprotectant (OP) and Antioxidant Levels

To quantify leaf contents of total soluble sugars (mg g^−1^ DW) and proline (μM g^−1^ DW), the procedures followed were those described in [117,118], respectively. In the homogenates of leaf tissues, ascorbate (AsA) and glutathione (GSH) were quantified (µM g^−1^ FW) applying the methods of [119,120], respectively.

The α-tocopherol (αToc) contents (µM g^−1^ DW) were obtained according to [121,122,123]. An extraction solvent (900 mL of C_10_H_22_O_2_, C_6_H_14_, and 100 mL of C_4_H_8_O_2_) was used, in which 0.02 g of C_15_H_24_O was dissolved. With R-Toc, 0.05 g/0.1 mL C_6_H_14_ functioned as a stock solution for a number of standards (20–200 µg mL^−1^). Sample preparation and saponification were done. Leaf tissue slices were dried at 40 °C, homogenized, and suspended in water, and 21 g of KOH dissolved in 100 mL of ethyl alcohol was added. For the test, 250 mg of ascorbate was added. Forty-min saponification was performed at 80 °C, and then samples were cooled directly. Using distilled water, the ratio of ethyl alcohol to water was adjusted to 0.3. N-hexane and ethyl acetate were added at 9 mL: 1 mL. Extraction was performed 3 times for the mixtures and water was used to wash out the combination. After filtering the organic phases, evaporation was performed to dryness. The residues were stored at −20 °C after dissolving by C_6_H_14_ (HPLC grade). αToc was estimated via a mobile phase by HPLC system.

### 4.11. Assay of Antioxidant Enzyme Activities and Enzyme Gene Expression

Polyvinylpyrrolidone (PVP) (1%)-containing K-phosphate buffer (ice-cold, pH 7.0) was utilized to extract leaf enzymes. Under 4 °C, the homogenates were centrifuged at 12,000 rpm for 15 min. The resulting enzyme extract was used to assay the enzyme activities, except for SOD. The superoxide dismutase (SOD, EC 1.15.1.1) activity was assessed using the nitro blue tetrazolium (NBT) method of Giannopolitis and Ries [124], with units defined as the quantity of enzyme required to prevent 50% of the NBT degradation rate at 560 nm. The [125,126,127,128] procedures were applied to assay CAT, APX, and GR activities (µM H_2_O_2_ min^−1^ g^−1^ protein), as well as SOD activity (Unit mg^−1^ protein), respectively. The procedures of [129] were applied to evaluate leaf soluble protein content (g kg^−1^ DW).

From leaf samples, total RNA was isolated utilizing a RNeasy Mini Kit (Qiagen GmbH, Stockach, Germany). Then, the posterior cDNA was synthesized using a RevertAid H Minus First Strand cDNA Synthesis Kit (Fermentas GmbH, St. Leon-Rot, Germany). Appendix A displays primer sequences for qRT-PCR of enzyme genes in tomato plants. By applying the directives of the industrialist of iQ SYBR Green Supermix (Bio-Rad, Hercules, CA, USA), qRT-PCR was analyzed. Two actin genes were used as a reference for the normalization of qPCR data. The efficiency of the reactions was calculated with LinRegPCR Software (version 11.0, download: http://LinRegPCR.HFRC.nl, accessed on 5 April 2023) [130]. In the analysis, qPCR raw data were utilized. Baseline corrections were made with the baseline trend depending on early cycle choices or using the developed algorithm. For each single sample, PCR efficiency was derived from the slope of the regression line fitted to a subset of baseline-corrected data points in the log-linear phase using LinRegPCR. Posteriorly, the formula of [131] was applied. With simple mathematics, gene-expression ratios can be calculated (ratio = N_0,target_/N_0,reference_) besides fold difference in gene expression between experimental conditions (fold = ratio_experiment_/ratio_control_).

### 4.12. Determination of Phytohormone Contents 

Indole-3-acetic acid (IAA), gibberellic acid (GA_3_), and cytokinin (CK) profiling was evaluated by applying the gas chromatography–mass spectrometry (GC-MS) system of [132,133]. An amount of 100 mg of fresh leaf sample was extracted in 2 mL of CH_3_OH:H_2_O:6 N HCl (ice-cold, 80:19.9:0.1, *v*/*v*/*v*). Under 4 °C, a 5-min centrifugation (25,000 rpm) of the resulting extract was performed. After collecting, the supernatant reached 50 μL using an N stream and then was freeze-stored (−80 °C) up to use. After extraction, the HPLC system was utilized to evaluate ABA level [134]. 

### 4.13. Statistical Analysis Tests 

Two-way ANOVA for the split-plot design (two factors were assigned; watering levels (100, 80, and 60% of ETc) were the first factor intended for main plots, while the NABF was the second factor intended for sub-plots) was applied to the statistical setup of the obtained data. Before beginning the analysis, all data were tested for the homogeneity of error variances [135] and normality distribution [136]. Statistically significant differences between means were assessed at a 5% and 1% probability level (*p* ≤ 0.05 and 0.01) using Tukey’s HSD (honestly significant difference) test. The GenStat 17th Ed. (VSN International Ltd., Hemel Hempstead, UK) software was applied for statistical analysis.

## 5. Conclusions

Drought-exposed tomato plants showed a diminution in growth, yield, fruit quality, photosynthetic efficiency, leaf integrity, and different leaf nutrient contents. However, the exogenous application of NABF noticeably promoted the levels of different antioxidants, osmoprotectants, and hormones, while electrolyte leakage, lipid peroxidation, hydrogen peroxide, and superoxide were diminished. Foliar applied NABF promoted the integrity of nutritional and hormonal balance, leaf tissues, photosynthesis efficiency, and photosynthesis-related pigments. This led to promoted plant growth, yield, fruit quality, and water use efficiency of drought-stressed tomato plants. Therefore, NABF can be applied as an affirmative strategy to overcome the damage of drought stress in tomato plants for sustainable tomato production in dry environments. Under the conditions of newly reclaimed saline soils, the practice of “drought stress × NABF” was explored as a very promising policy to maximize water use efficiency. To precisely determine the enhancement mechanisms of NABF, further research and analysis are needed, making it a sought-after commercial compound in the agricultural market.

## Figures and Tables

**Figure 1 plants-12-03407-f001:**
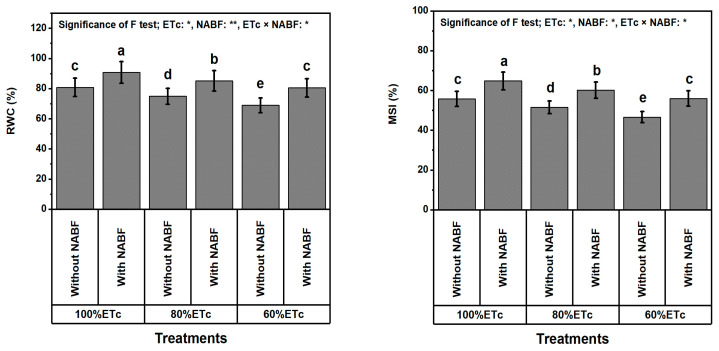
Impact of foliar supplementation with a novel nutrient- and antioxidant-based formulation (NABF) on leaf tissue cell integrity of tomato plants that experienced full watering and drought stress (80 or 60% of crop evapotranspiration; ETc). ^(^*^)^ or ^(^**^)^ signalizes differences at *p* ≤ 0.01 or 0.05 level of probability, respectively. In each plot, based on the LSD test (*p* ≤ 0.05), columns (±SE bar) with different letters are significantly different (n = 3). RWC, relative water content; MSI, membrane stability index.

**Figure 2 plants-12-03407-f002:**
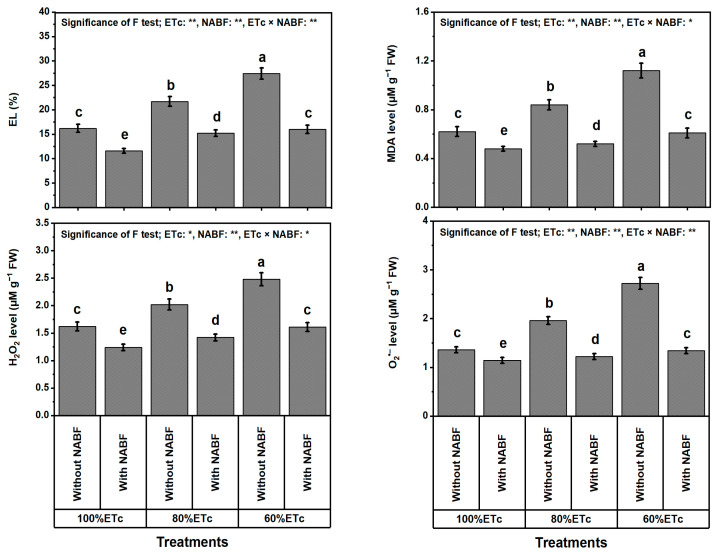
Impact of foliar supplementation with a novel nutrient- and antioxidant-based formulation (NABF) on leaf oxidant levels and their damage in terms of ionic leakage and lipid peroxidation (assessed as malondialdehyde content) of tomato plants that experienced full watering and drought stress (80 or 60% of crop evapotranspiration; ETc). ^(^*^)^ or ^(^**^)^ signalizes differences at *p* ≤ 0.01 or 0.05 level of probability, respectively. In each plot, based on the LSD test (*p* ≤ 0.05), columns (±SE bar) with different letters are significantly different (n = 3). FW, fresh weight; EL, electrolyte leakage; MDA, malondialdehyde; H_2_O_2_, hydrogen peroxide; O_2_^•−^, superoxide.

**Figure 3 plants-12-03407-f003:**
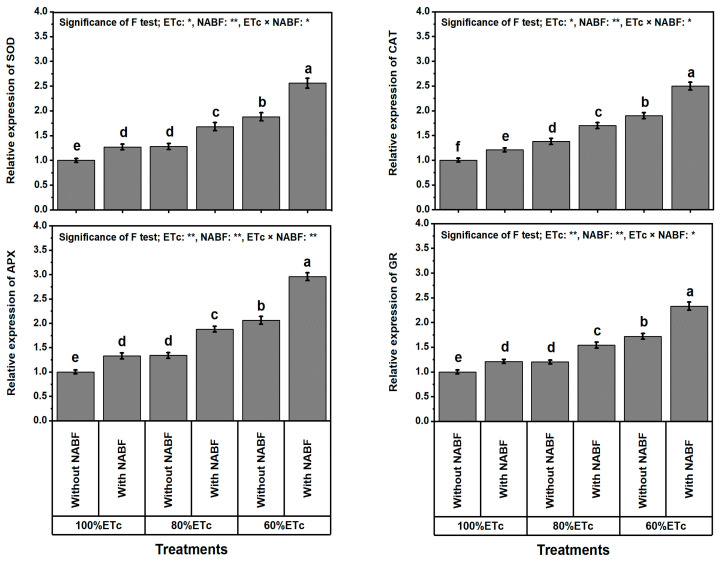
Impact of foliar supplementation with a novel nutrient- and antioxidant-based formulation (NABF) on transcript levels of antioxidant enzyme encoding genes of tomato plants that experienced full watering and drought stress (80 or 60% of crop evapotranspiration; ETc). ^(^*^)^ or ^(^**^)^ signalizes differences at *p* ≤ 0.01 or 0.05 level of probability, respectively. In each plot, based on the LSD test (*p* ≤ 0.05), columns (±SE bar) with different letters are significantly different (n = 3). SOD, superoxide dismutase; CAT, catalase; APX, ascorbate peroxidase; GR, glutathione reductase.

**Figure 4 plants-12-03407-f004:**
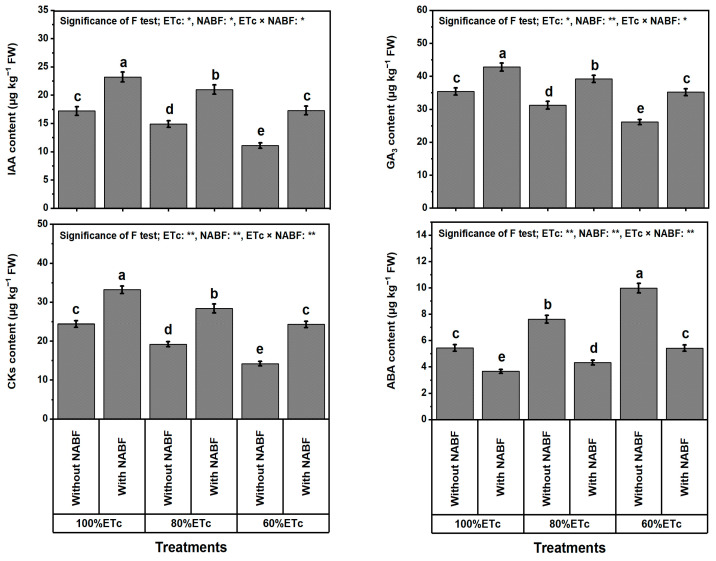
Impact of foliar supplementation with a novel nutrient- and antioxidant-based formulation (NABF) on hormonal contents of tomato plants that experienced full watering and drought stress (80 or 60% of crop evapotranspiration; ETc). ^(^*^)^ or ^(^**^)^ signalizes differences at *p* ≤ 0.01 or 0.05 level of probability, respectively. In each plot, based on the LSD test (*p* ≤ 0.05), columns (±SE bar) with different letters are significantly different (n = 3). FW, fresh weight; IAA, indole-3-acetic acid; GA_3_, gibberellic acid; CKs, cytokinins; ABA, abscisic acid.

**Table 1 plants-12-03407-t001:** Impact of foliar supplementation with a novel nutrient- and antioxidant-based formulation (NABF) on performance of tomato plants grown under full watering and drought stress (80 or 60% of crop evapotranspiration; ETc).

Treatments		PLN	PLA (m^2^)	PSDW (g)	PFW (kg)	PFN	HFY (ton)	WUE (kg Fruits m^−3^ Water)
ETc	NABF							
100%	Without	59.1 ± 4.80 c	5.19 ± 0.38 c	63.4 ± 4.8 c	5.18 ± 0.42 c	32.4 ± 3.10 c	69.1 ± 5.2 c	12.14 ± 1.7 d
	With	79.0 ± 6.21 a	6.98 ± 0.50 a	99.3 ± 7.2 a	9.30 ± 0.82 a	54.1 ± 4.14 a	121.2 ± 9.8 a	21.29 ± 1.9 a
80%	Without	45.3 ± 3.44 d	4.01 ± 0.31 d	49.8 ± 3.5 d	2.76 ± 0.11 d	19.7 ± 1.35 d	36.8 ± 2.41 d	8.08 ± 1.2 e
	With	63.4 ± 5.30 b	5.54 ± 0.38 b	71.2 ± 5.7 b	5.64 ± 0.44 b	36.1 ± 3.01 b	76.2 ± 6.3 b	16.73 ± 1.6 c
60%	Without	32.1 ± 2.82 e	2.70 ± 0.13 e	33.2 ± 2.8 e	0.81 ± 0.08 e	10.1 ± 0.91 e	20.2 ± 0.8 e	5.91 ± 0.71 f
	With	58.4 ± 4.72 c	5.05 ± 0.34 c	63.1 ± 4.9 c	5.10 ± 0.40 c	31.7 ± 2.51 c	68.4 ± 5.1 c	20.02 ± 1.3 b
Significance of F test:								
ETc		*	*	**	*	**	*	**
NABF		**	**	**	**	**	**	**
ETc × NABF		*	*	**	*	**	*	**

^(^*^)^ or ^(^**^)^ signalizes differences at *p* ≤ 0.01 or 0.05 level of probability, respectively. In each column, based on the LSD test (*p* ≤ 0.05), means ± SE with different letters are significantly different (n = 3). PLN, plant leaf number; PLA, plant leaf area; PSDW, dry weight of plant shoot; PFW, plant fruit weight; PFN, plant fruit number; HFY, fruit yield per hectare; and WUE, water use efficiency.

**Table 2 plants-12-03407-t002:** Impact of foliar supplementation with a novel nutrient- and antioxidant-based formulation (NABF) on fruit quality-related traits of tomato plants cultivated under full watering and drought stress (80 or 60% of crop evapotranspiration; ETc).

Treatments		Vit C (mg 100 g^−1^ Fruit FW)	TSS (°Brix)	TA (% Citric Acid)	Lycopene (mg 100 g^−1^ Fruit FW)	β-Car (mg 100 g^−1^ Fruit FW)	Firmness (kg cm^−1^)	Selenium (mg kg^−1^ FW)	Iodine (mg kg^−1^ FW)
ETc	NABF								
100%	Without	5.86 ± 0.15 d	4.89 ± 0.38 d	0.39 ± 0.09 e	4.68 ± 0.19 e	2.02 ± 0.07 e	4.60 ± 0.24 c	3.02 ± 0.08 c	1.18 ± 0.04 c
	With	6.39 ± 0.18 c	5.28 ± 0.44 c	0.45 ± 0.12 d	4.99 ± 0.21 d	2.25 ± 0.09 d	4.99 ± 0.27 a	3.62 ± 0.10 a	1.48 ± 0.05 a
80%	Without	6.41 ± 0.17 c	5.26 ± 0.46 c	0.50 ± 0.14 c	5.24 ± 0.22 c	2.46 ± 0.10 c	4.32 ± 0.22 d	2.52 ± 0.06 d	1.02 ± 0.03 d
	With	7.09 ± 0.20 b	5.69 ± 0.49 b	0.59 ± 0.18 b	5.72 ± 0.25 b	2.78 ± 0.11 b	4.74 ± 0.25 b	3.34 ± 0.07 b	1.32 ± 0.03 b
60%	Without	7.11 ± 0.20 b	5.70 ± 0.50 b	0.58 ± 0.19 b	5.78 ± 0.26 b	2.80 ± 0.11 b	4.10 ± 0.18 e	2.12 ± 0.05 e	0.82 ± 0.02 e
	With	7.94 ± 0.24 a	6.27 ± 0.54 a	0.70 ± 0.21 a	6.38 ± 0.30 a	3.22 ± 0.12 a	4.61 ± 0.25 c	3.01 ± 0.08 c	1.19 ± 0.04 c
Significance of F test:									
ETc		*	*	*	*	**	*	**	**
NABF		*	**	**	*	**	*	**	**
ETc × NABF		*	*	*	*	**	*	**	**

^(^*^)^ or ^(^**^)^ signalizes differences at *p* ≤ 0.01 or 0.05 level of probability, respectively. In each column, based on the LSD test (*p* ≤ 0.05), means ± SE with different letters are significantly different (n = 3). Vit C, vitamin C; TSS, total soluble solids; TA, titratable acidity; β-Car, β-carotene.

**Table 3 plants-12-03407-t003:** Impact of foliar supplementation with a novel nutrient- and antioxidant-based formulation (NABF) on photosynthetic efficiency-related traits of tomato plants cultivated under full watering and drought stress (80 or 60% of crop evapotranspiration; ETc).

Treatments		SPAD Value	Chlorophyll	Carotenoids	Fv/Fm	PI	PhAc
ETc	NABF		mg g^−1^ FW			%	
100%	Without	47.6 ± 2.02 c	3.22 ± 0.16 c	0.36 ± 0.02 c	0.79 ± 0.03 c	16.8 ± 0.32 c	49.1 ± 1.03 c
	With	56.8 ± 2.12 a	3.52 ± 0.18 a	0.47 ± 0.03 a	0.88 ± 0.04 a	19.2 ± 0.38 a	58.9 ± 1.14 a
80%	Without	42.1 ± 1.94 d	2.87 ± 0.12 d	0.32 ± 0.02 d	0.73 ± 0.02 d	14.9 ± 0.29 d	44.1 ± 0.98 d
	With	52.8 ± 2.02 b	3.36 ± 0.15 b	0.43 ± 0.03 b	0.84 ± 0.03 b	17.9 ± 0.35 b	54.6 ± 1.04 b
60%	Without	29.8 ± 1.11 e	2.11 ± 0.10 e	0.24 ± 0.01 e	0.68 ± 0.02 e	12.1 ± 0.22 e	34.4 ± 0.82 e
	With	47.3 ± 2.01 c	3.23 ± 0.17 c	0.36 ± 0.02 c	0.79 ± 0.03 c	16.6 ± 0.33 c	48.9 ± 1.06 c
Significance of F test:							
ETc		*	**	**	*	*	*
NABF		**	**	**	*	**	*
ETc × NABF		*	**	**	*	*	*

^(^*^)^ or ^(^**^)^ signalizes differences at *p* ≤ 0.01 or 0.05 level of probability, respectively. In each column, based on the LSD test (*p* ≤ 0.05), means ± SE with different letters are significantly different (n = 3). SPAD, Soil Plant Analysis Development (specifies the relative content of chlorophyll as a leaf greenness); Fv/Fm, quantum efficiency of PSII; PI, performance index; PhAc, photochemical activity.

**Table 4 plants-12-03407-t004:** Impact of foliar supplementation with a novel nutrient- and antioxidant-based formulation (NABF) on leaf macronutrient contents of tomato plants cultivated under full watering and drought stress (80 or 60% of crop evapotranspiration; ETc).

Treatments		Nitrogen	Phosphorus	Potassium	Calcium	Magnesium	Sulfur
ETc	NABF	mg g^−1^ DW					
100%	Without	18.6 ± 0.42 c	1.99 ± 0.05 c	18.1 ± 0.39 c	6.88 ± 0.16 c	1.69 ± 0.04 c	0.84 ± 0.02 c
	With	26.4 ± 0.62 a	2.52 ± 0.08 a	25.2 ± 0.61 a	8.78 ± 0.18 a	2.14 ± 0.06 a	1.12 ± 0.03 a
80%	Without	14.2 ± 0.33 d	1.73 ± 0.04 d	15.3 ± 0.32 d	5.42 ± 0.14 d	1.46 ± 0.03 d	0.62 ± 0.02 d
	With	21.2 ± 0.53 b	2.28 ± 0.07 b	22.8 ± 0.50 b	8.15 ± 0.15 b	1.94 ± 0.05 b	0.98 ± 0.03 b
60%	Without	11.3 ± 0.28 e	1.45 ± 0.03 e	12.1 ± 0.29 e	3.94 ± 0.12 e	1.12 ± 0.03 e	0.42 ± 0.01 e
	With	18.8 ± 0.40 c	1.98 ± 0.05 c	18.2 ± 0.41 c	6.90 ± 0.17 c	1.68 ± 0.04 c	0.83 ± 0.02 c
Significance of F test:							
ETc		**	*	**	**	*	**
NABF		*	**	**	**	**	**
ETc × NABF		*	*	*	**	*	**

^(^*^)^ or ^(^**^)^ signalizes differences at *p* ≤ 0.01 or 0.05 level of probability, respectively. In each column, based on the LSD test (*p* ≤ 0.05), means ± SE with different letters are significantly different (n = 3).

**Table 5 plants-12-03407-t005:** Impact of foliar supplementation with a novel nutrient- and antioxidant-based formulation (NABF) on leaf micronutrient contents of tomato plants grown under full watering and drought stress (80 or 60% of crop evapotranspiration; ETc).

Treatments		Iron	Manganese	Zinc	Copper	Selenium	Iodine
ETc	NABF	mg kg^−1^ DW					
100%	Without	702 ± 16.8 c	528 ± 13.2 c	304 ± 7.2 c	240 ± 6.1 c	18.2 ± 0.46 c	3.58 ± 0.12 c
	With	844 ± 18.8 a	632 ± 14.8 a	381 ± 8.4 a	292 ± 7.2 a	22.7 ± 0.61 a	4.49 ± 0.15 a
80%	Without	582 ± 14.5 d	466 ± 12.6 d	262 ± 6.4 d	212 ± 5.4 d	15.4 ± 0.34 d	3.06 ± 0.13 d
	With	784 ± 15.4 b	584 ± 14.0 b	348 ± 7.8 b	268 ± 6.7 b	20.1 ± 0.41 b	4.01 ± 0.14 b
60%	Without	440 ± 12.9 e	392 ± 11.1 e	212 ± 5.3 e	182 ± 4.6 e	12.5 ± 0.29 e	2.40 ± 0.07 e
	With	698 ± 16.7 c	526 ± 13.0 c	301 ± 7.1 c	242 ± 6.3 c	18.1 ± 0.48 c	3.60 ± 0.13 c
Significance of F test:							
ETc		**	*	**	**	*	**
NABF		*	**	**	**	**	**
ETc × NABF		*	*	*	**	*	**

^(^*^)^ or ^(^**^)^ signalizes differences at *p* ≤ 0.01 or 0.05 level of probability, respectively. In each column, based on the LSD test (*p* ≤ 0.05), means ± SE with different letters are significantly different (n = 3).

**Table 6 plants-12-03407-t006:** Impact of foliar supplementation with a novel nutrient- and antioxidant-based formulation (NABF) on osmoprotectant and antioxidant contents of tomato plants grown under full watering and drought stress (80 or 60% of crop evapotranspiration; ETc).

Treatments		S. Sugar Content	Free Proline	AsA Content	GSH Content	αToc
ETc	NABF	(mg g^−1^ DW)	(mM g^−1^ DW)	(µM g^−1^ FW)		(µM g^−1^ DW)
100%	Without	21.6 ± 0.11 d	0.12 ± 0.003 f	1.70 ± 0.03 d	0.80 ± 0.02 f	2.28 ± 0.05 d
	With	24.8 ± 0.13 c	0.14 ± 0.004 e	1.88 ± 0.04 c	0.86 ± 0.02 e	2.52 ± 0.06 c
80%	Without	25.2 ± 0.13 c	0.17 ± 0.005 d	1.89 ± 0.04 c	0.93 ± 0.03 d	2.54 ± 0.06 c
	With	29.2 ± 0.15 b	0.20 ± 0.006 c	2.22 ± 0.05 b	1.04 ± 0.03 c	2.88 ± 0.07 b
60%	Without	29.0 ± 0.15 b	0.24 ± 0.007 b	2.24 ± 0.05 b	1.66 ± 0.04 b	2.91 ± 0.07 b
	With	35.4 ± 0.18 a	0.29 ± 0.008 a	2.74 ± 0.06 a	1.98 ± 0.04 a	3.42 ± 0.08 a
Significance of F test:						
ETc		*	*	**	**	*
NABF		**	**	**	**	**
ETc × NABF		*	*	**	**	*

^(^*^)^ or ^(^**^)^ signalizes differences at *p* ≤ 0.01 or 0.05 level of probability, respectively. In each column, based on the LSD test (*p* ≤ 0.05), means ± SE with different letters are significantly different. S. sugars, soluble sugars; AsA, ascorbate; GSH, glutathione; αToc, α-Tocopherol.

**Table 7 plants-12-03407-t007:** Impact of foliar supplementation with a novel nutrient- and antioxidant-based formulation (NABF) on enzyme activities and soluble protein content of tomato plants grown under full watering and drought stress (80 or 60% of crop evapotranspiration; ETc).

Treatments		SOD Activity	CAT Activity	APX Activity	GR Activity	SPC
ETc	NABF	Unit g^−1^ Protein	µmol H_2_O_2_ min^−1^ g^−1^ Protein			g kg^−1^ DW
100%	Without	36.9 ± 0.85 e	17.1 ± 0.38 f	23.2 ± 0.42 e	16.0 ± 0.24 d	232 ± 5.2 c
	With	41.2 ± 0.92 d	18.9 ± 0.42 e	25.9 ± 0.48 d	17.9 ± 0.28 c	282 ± 6.7 a
80%	Without	41.9 ± 0.93 d	21.7 ± 0.48 d	26.2 ± 0.49 d	18.2 ± 0.30 c	206 ± 4.8 d
	With	47.8 ± 0.97 c	26.2 ± 0.50 c	30.6 ± 0.55 c	22.2 ± 0.35 b	258 ± 5.8 b
60%	Without	54.4 ± 1.02 b	31.5 ± 0.55 b	36.2 ± 0.59 b	22.9 ± 0.36 b	175 ± 3.9 e
	With	65.8 ± 1.10 a	39.8 ± 0.63 a	44.4 ± 0.70 a	29.2 ± 0.42 a	230 ± 5.1 c
Significance of F test:						
ETc		**	**	*	**	*
NABF		**	**	*	**	*
ETc × NABF		*	**	*	**	*

^(^*^)^ or ^(^**^)^ signalizes differences at *p* ≤ 0.01 or 0.05 level of probability, respectively. In each column, based on the LSD test (*p* ≤ 0.05), means ± SE with different letters are significantly different (n = 3). SOD, superoxide dismutase; CAT, catalase; APX, ascorbate peroxidase; GR, glutathione reductase; SPC, soluble protein.

**Table 8 plants-12-03407-t008:** Physicochemical properties of the tested soil.

Soil Layer (cm)	Particle Size Distribution										
	Clay (%)			Silt (%)			Sand (%)			Texture	
	25.7 ± 0.62			32.7 ± 0.45			41.6 ± 0.52			Loam	
0–30	BD (g cm^−3^)		K_sat_ (cm h^−1^)		FC (%)			WP (%)			AW (%)
	1.58 ± 0.03		1.89 ± 0.02		24.2 ± 0.34			11.4 ± 0.14			12.8 ± 0.11
pH		7.88 ± 0.03				N (mg kg^−1^ soil)			46.8 ± 0.87		
ECe (dS m^−1^)		10.2 ± 0.05				P (mg kg^−1^ soil)			5.12 ± 0.12		
CEC (cmol kg^−1^)		12.9 ± 0.06				K (mg kg^−1^ soil)			37.2 ± 0.67		
CaCO_3_ (%)		2.27 ± 0.04				Fe (mg kg^−1^ soil)			2.92 ± 0.03		
OM (%)		0.98 ± 0.01				Mn (mg kg^−1^ soil)			1.62 ± 0.03		
ESP		14.2 ± 0.11				Zn (mg kg^−1^ soil)			1.34 ± 0.02		
SAR		10.4 ± 0.06				Cu (mg kg^−1^ soil)			0.72 ± 0.01		

Abbreviations: BD, bulk density; K_sat_, hydraulic conductivity; FC, field capacity; WP, wilting point; AW, available water; OM, organic matter; ESP, exchangeable Na^+^ percentage; SAR, Na^+^ adsorption ratio. Values are means ± SE (n = 3).

**Table 9 plants-12-03407-t009:** Composition of a novel nutrient- and antioxidants-based formulation (NABF) utilized in this study.

The Component	% *w*/*w*	Molecular Formula	ppm	mM L^−1^	MW
Urea (N)	7.2	CO(NH_2_)_2_	180	2.99	60.06
MAP (P)	9.2	NH_4_H_2_PO_4_	230	1.99	115.03
Potassium sulfate (K)	20.9	K_2_SO_4_	523	3.00	174.26
Magnesium sulfate (Mg)	7.2	MgSO_4_	180	1.49	120.37
Calcium nitrate (Ca)	13.1	Ca(NO_3_)_2_	328	1.99	164.09
Sodium selenate (Se)	3.0	Na_2_SeO_3_	75	0.43	172.94
Potassium iodide (I)	2.0	KI	50	0.30	166.00
Ascorbic acid (AsA)	7.0	C_6_H_8_O_6_	175	0.99	176.12
Glutathione (GSH)	6.1	C_10_H_17_N_3_O_6_S	152	0.49	307.32
Salicylic acid (SA)	3.3	C_19_H_19_N_7_O_6_	82	0.59	138.12
Citric acid (CA)	15.4	C_6_H_8_O_7_	385	2.00	192.12
EDTA	5.6	C_10_H_16_N_2_O_8_	140	0.48	292.24

MW, molecular weight; MAP, mono-ammonium phosphate; EDTA, ethylenediaminetetraacetic acid. All components of NABF were obtained from Sigma-Aldrich (Sigma-Aldrich Co. LLC, St. Louis, MO, USA).

**Table 10 plants-12-03407-t010:** Average 2-season weather data of the experimental area.

Month	Day °C	Night °C	ARH (%)	AWS (km h^−1^)	AP (mm d^−1^)
2020 and 2021 Seasons, Respectively					
February	26.5 and 27.0	4.14 and 3.16	64.8 and 56.4	2.24 and 2.45	1.91 and 0.00
March	30.7 and 34.4	4.87 and 5.13	57.6 and 53.6	2.76 and 2.86	0.41 and 0.51
April	34.5 and 40.4	9.34 and 5.62	49.2 and 41.1	2.92 and 3.26	0.05 and 0.02
May	44.2 and 43.5	12.0 and 15.7	39.9 and 30.6	3.58 and 3.50	0.00 and 0.00
June	42.9 and 41.0	17.3 and 16.3	35.0 and 30.2	3.59 and 3.97	0.00 and 0.00

Day °C, average day temperature; Night °C, average night temperature; ARH, average relative humidity; AWS, average wind speed; AP, average precipitation.

**Table 11 plants-12-03407-t011:** Irrigation water analysis.

Ion Concentrations (meq L^−1^)								EC (dS m^−1^)	pH	SAR
CO_3_^2−^	HCO_3_^−^	SO_4_^2−^	Cl^−^	Mg^2+^	Ca^2+^	K^+^	Na^+^			
0.00 ± 0.00	2.10 ± 0.11	3.27 ± 0.19	12.1 ± 0.46	1.76 ± 0.16	5.38 ± 0.34	1.42 ± 0.14	6.16 ± 0.64	1.71 ± 0.21	7.51 ± 0.71	2.94 ± 0.24

EC, electrical conductivity; SAR, Na^+^ adsorption ratio. Values are means ± SE (n = 3).

## Data Availability

The data presented in this study are available upon request from the corresponding author.

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
