# Peer review of "A Novel Nutrient- and Antioxidant-Based Formulation Can Sustain Tomato Production under Full Watering and Drought Stress in Saline Soil"

_plants, 2023, doi:10.3390/plants12193407_

Round 1
Reviewer 1 Report
In my opinion, the investigation of the efficacy of the NABF product has been carried out correctly. The manuscript is well written and detailed, the conclusions agreeing with the results presented. In the attached document I indicate some aspects that should be clarified prior to its publication, and others as suggestions with the idea of improving certain aspects. By conducting the experiment in a saline soil, apparently a reclaimed soil, I think more emphasis should be placed on including the benefit of this product in saline conditions. It would be desirable if more information could be given on what treatments this soil has undergone for its recovery. There are remarkable aspects such as the fact that the levels of macronutrients in the leaves are not very high, and yet the levels of micronutrients are very high. Nor is it detailed what fertilization has been applied to the crop, or under what conditions or phenology. Although the results are eloquent, perhaps the photosynthetic parameters, instead of being calculated, could have been measured with a photosynthesis analyzer. I indicate it, since the work seems quite complete to me, for the purposes of future studies. In the attached document I indicate details of those aspects that I have thought convenient to highlight.

English is fine to me.
Author Response
Response to Reviewer 1#
Dear Reviewer,
Thank you for your notes. We highly appreciate your insightful and helpful comments on our manuscript. We carefully improved the manuscript and we hope that we satisfactorily referred to your comments. For clarity, we provided a point-by-point response to your comments. We highlighted the revised sections with a yellow color.
Point 1: In the title maybe you should also indicate in salt conditions
Response 1: Thank you for your note. The title has been changed (Lines 2-4).
Point 2: Abstract, if possible, shorten to around 200-250 words. Include in the Abstract only the complete name (in case it is not repeated in the Abstract). Include the complete name in Material and Methods. Same comment for the rest of parameters.
Response 2: The abstract was shortened to 275 words (The minimum we could reach in terms of reducing the number of words due to the large number of results tested in this manuscript) and the complete names are included (Lines 20-39).
Point 3: In the keywords add salt stress
Response 3. Done as requested (Line 40).
Point 4: In line 89 please add author name before et al. [30]. If possible, place at the end of the sentence.
Response 4. Done as requested (Line 72).
Point 5: In line 110, To place at the end of the sentence.
Response 5: Done as requested (Lines 80 and 99).
Point 6: According to Plants, place this point after Discussion.
Response 6: Done as requested.
Point 7: You can include also deviations in data
Response 7: The metrological station gave only one reading for every variable (i.e., average day temperature, average relative humidity, average wind speed, and average precipitation), so the deviations cannot be included.
Point 8: Please, describe briefly the methods used to analyze soil physicochemical properties.
Response 8: Done (Lines 520-540).
Point 9: In Table 2, please, include deviations and number of samples taken to establish means
Response 9: Done (please see Table 9, Lines 544-547).
Point 10: In line 147, please, include procedence and country.
Response 10: done as requested (Lines 550-551).
Point 11: Please, include the planting frame
Response 11: The experiments were designated as a split-plot arrangement for six treatments with 3 replicates, and each replicate was represented by a row 15 m long and 1.5 m wide, which contained 30 plants (Lines 553-555).
Point 12: Is this formulation patented? In case, include details of patent.
Response 12: We are now working on the procedures to register the formulation as a patent.
Pont 13: Please, include also the total dose applied per plant in the entire crop cycle.
Response 13: The total dose applied per plant was 0.45g (Lines 576-577).
Point 14: Can you include the parameters you studied?
Response 14: plant leaf number, plant leaf area, dry weight of plant shoot (Line 562).
Point 15: Please, include in brackets the plant's age.
Response 15: Done, 2 WAT (Line 574).
Point 16: Were these treatments coincided with certain phenological stages?
Response 16: The vegetative and flowering stages (Line 575).
Point 17: Please, include it (fertilization program) also here with the modifications. Also the total amount or doses of fertilizers applied to the crop.
Response17: The fertilization program has been added (Lines 581-592).
Point 18: Can you include briefly methods of analysis (water)?
Response 18: Irrigation water samples were analyzed in a special laboratory at the National Research Center, Egypt and we sent a request for analysis methods, but we did not receive a response from them until the date of submitting the revisions.
Point 19: Can you include deviations and number of repetitions?
Response 19: Done (please see Table 9, lines 544-547).
Point 20: Maybe better to include in point 2.4 Plant sampling and processing (for example)
Response 20: Done, please see lines 617-624.
Point 21: Please, include also the complete name (TMAH).
Response 21: Tetra methyl ammonium hydroxide (Lines 632-633).
Point 22: Please, explain ([prEN 15111: R2-P5-F01 2006 (E)]).
Response 22: Foodstuffs – Determination of trace elements – Determination of iodine by ICP-MS (inductively coupled plasma mass spectrometry). Polish Committee of Standardization. (In Polish) (Lines 634-635).
Point 23: Please, include the meaning of each variable [Fv/Fm = (Fm ‒ F0)/Fm].
Response 23: Fv is the variable fluorescence, Fm is the maximum fluorescence, and F0 is the basal fluorescence (Lines 654-656).
Point 24: Include here the complete name, and in Results include only the abbreviation. In legends write also the complete name.
Response 24: Done, (Line 670).
Point 25: In line 688, Please include normal conditions.
Response25: done see line 675.
Point26: Please, include details for SOD.
Response26: Done see lines 704-707.
Point 27: What did you consider as an Unit?
Response 27: An activity unit is an approved unit of the method of determination.
Point 28: Please include details of producer and country.
Response 28: Included (Lines 717-725).
Point 29: Can you include here to facilitate reading? (the formula).
Response 29: Included (Lines 717-725).
Point30: Please, include also the complete name (IAA, GA3, and CKs)
Response30: Done, (Line 727).
Point31: Complete name (GC-MS)
Response31: Done, see line 728.
Point 32: Did you perform Two-way ANOVA? please, explain including details of factors.
Response 32: Yes, two-way ANOVA was performed (please see Lines 735-737).
Point 33: Please, revise data levels of Fe, Cu, Mn and Zn because seems too high. Also for controls. Soil levels were poor on Fe and Cu; and, poor to medium for Mn and Zn.
Response33: revised and it is correct (mg/kg)
Point 34: Figure 1. Please, revise format for all Figures.
Response 34: The format of all figures has been revised.
Point 35: Please, include in the legend explanations for Significance F test for each variable and interaction. Also for the rest of Figures.
Response 35: Just as the significance of F test is present at the bottom of each column in the tables, it is also present at the top of each Figure unit.
Point 36: Please, include the repetitions (n =). Also for the rest of Figures.
Response 36: Done, (n=3)
Point 37: Don't affirm. As these aspects didn't studied
Response 37: Actually we studied photosynthetic efficiency (Fv/Fm and PI) and PhAc; photochemical activity. Please see Table 7.
Point 38: Can you include some examples in tomato? Also for the rest of the examples.
Response 38: Done.
Point 39: "probably mediated" because you didn't study these physiological mechanisms.
Response 39: Done.
Point 40: Maybe you can reduce a little the bibliography. Please, delete those with similar findings. Also in the rest of the manuscript.
Response 39: Done.
Point 40: Please, explain, if possible, why control plants, showed high levels of micronutrients.
Response 40: The contents of micronutrients are higher in treated plants with NABF than untreated pants (control), please see Table 9.
Point 41: I think this paragraph is redundant.
Response 41: The paragraph was deleted.
Point 41: In my opinion, "similar" would be when the work you cited is at least in the same culture. If you don't find examples in tomato, then you can indicate, "in wheat was found...". I think you should revise the rest of examples cited in the Discussion.
Response41: Done.
Point 42: The fact that you carried out the experiment in a salt soil, makes that more important to this type of stress should be highlighted.
Response 42: We would like to thank Reviewer #1 for this note.
Point 43: Please, rewrite lines 752-754
Response 43: Done.
Point 44: References should be shortened most focused on tomato.
Response 44: Done.
Point 45: Please, revise the format of references in the template.
Response 45: All references were entirely according to as the guideline of plants MDPI Journal.

Reviewer 2 Report
plants-2575353-peer-review-v1
Article:
A novel nutrient and antioxidant-based formulation can sustain tomato production under water shortfall stress conditions
Abstracts:
Abstract must be short, for example the last paragraph, which includes the phrase “saving water,” which is not the primary goal of the manuscript.
Keywords:
Remove Oxidative stress and Nutrient and hormonal ho-45 meostasis .
"Tomato growth and yield quality" correct "Tomato, Yield"
Introduction:
Line 56-58: :and Egypt being 56 the top five producing countries [5]. In Egypt, tomatoes rank first among all vegetable 57 crops in cultivated areas and is mainly utilized for local consumption and partially for 58 exportation" remove this paragraph.
In this paragraph, Throughout its life cycle, tomato can experience negative effects when exposed to environmental challenges [8]. Moreover, climate change, nutrient deficiency, heavy metal, high carbonate content, and other soil conditions direct affect agriculture, especially in the Mediterranean [9–11]. In both dry and semiarid situations, soil salinity remains a hazardous factor that limits agricultural production. It has an impact not only on food production but also on arable land all across the world. By 2050, it is estimated that over 20% of irrigated land will be off-duty and 50% of cultivable land will be affected by salinity [12]. What does salinity have to do with this manuscript?, there is no need for this paragraph.
The authors mentioned the name of Egypt a lot in the introduction. Please correct that.
Line 89: [30] reported that correct to Kumari et al [30]
Line 94: In this concern, [23,31–36], reported that "correct"
Line 126-128: According to [37], ??
Line 96: The ability of tomato plants to withstand WSS cir-127 cumstances was investigated by evaluating some physio-biochemical indices "Remove".
Materials and Methods:
Line 147-148: Forty-day-old tomato (Solanum lycopersicum L.) transplants (cultivar 320) were se-147 cured by the Nurseries of Agriculture Ministry, Cairo, Egypt " Needs rewriting"
Line 157-161: This concentration (2.5 g L−1) of the NABF was selected based on a preliminary study, in which five times foliar nourishment with 2.5 g NABF L−1 gave the best tomato growth results among several concentrations used (data not shown). Additionally, this formulation was selected from several formulations due to its best results (data not shown). The NABF composition is presented in Table 3. (It is required to clarify and review the data that is not displayed and add it to supplementary data).
Line 165: This formulation was used at a rate of 2.5 g L−1. (Remove)
Line 176: Also, the fertilization program detailed in [46] with some modifications. "Needs rewriting"
2.4. Plant Sampling 194
Twelve weeks after transplantation (WAT), plant samples were taken. Randomly, 9 plants were taken from each treatment (3 plants from each row/replicate) for growth traits, photosynthetic efficiency parameters, and nutrient contents. In the same way, another 9 plants were taken and assigned to enzyme assays, enzyme gene expression, and hormonal analysis. A third set of 9 plants was taken randomly from each treatment and assigned to further analyses (leaf integrity parameters, oxidative stress markers, osmoprotectants (OPS), and non-enzymatic antioxidants). Beginning at 12 WAT, tomato fruits were collected five times, every seven days, from the remaining plants in each treatment. The fruits were used to evaluate the fruit yield components and quality of fruits. (The paragraph needs more explanation of analysis methods and references).
2.5. Fruit Yield Characteristics, and Fruit Quality Traits:
Wrong title and what is mentioned after the title has nothing to do with the title. This indicates that the manuscript was not reviewed.
There are so many measurements in the manuscript, it is unbelievable.
NABF, Is it a commercial product?
Results:
The titles of all tables need to be revised, for example not mentioning the treatment 100%
Fig. 1,2 ,3,4. Need to improve.
Discussion: a good portion will do. But it can be shortened.
Conclusions: can be shortened.
References:
Scientific names are written in italics.
Extensive editing of English language required
Author Response
Response to Reviewer 2 # Comments
Dear Reviewer,
Thank you for your notes. We highly appreciate your insightful and helpful comments on our manuscript. We carefully improved the manuscript and we hope that we satisfactorily referred to your comments. For clarity, we provided a point-by-point response to your comments. We highlighted the revised sections with a blue color.
Point 1: Abstract must be short, for example the last paragraph, which includes the phrase “saving water,” which is not the primary goal of the manuscript.
Response 1 : The abstract was shorted to 275 words, and the last paragraph was modified (see the abstract section)
Point 2: Keywords: Remove Oxidative stress and Nutrient and hormonal homeostasis . "Tomato growth and yield quality" correct "Tomato, Yield"
Resonse 2: The Keywords has beean changed (Line 39).
Point 3: Line 56-58: :and Egypt being the top five producing countries [5]. In Egypt, tomatoes rank first among all vegetable crops in cultivated areas and is mainly utilized for local consumption and partially for exportation" remove this paragraph.
Resonse 3. The pargraph " In Egypt, tomatoes rank first among all vegetable crops in cultivated areas and is mainly utilized for local consumption and partially for exportation" was removed.
Point 4: In this paragraph, Throughout its life cycle, tomato can experience negative effects when exposed to environmental challenges [8]. Moreover, climate change, nutrient deficiency, heavy metal, high carbonate content, and other soil conditions direct affect agriculture, especially in the Mediterranean [9–11]. In both dry and semiarid situations, soil salinity remains a hazardous factor that limits agricultural production. It has an impact not only on food production but also on arable land all across the world. By 2050, it is estimated that over 20% of irrigated land will be off-duty and 50% of cultivable land will be affected by salinity [12]. What does salinity have to do with this manuscript?, there is no need for this paragraph.
Resonse 4. This paragraph has been removed (see the introduction section).
Point 5: The authors mentioned the name of Egypt a lot in the introduction. Please correct that
Response 5: The name of Egypt has been decreased.
Point 6: Line 89: [30] reported that correct to Kumari et al [30]
Response 5: corrected (Line 72).
Point 7: Line 94: In this concern, [23,31–36], reported that "correct"
Response 7: corrected, see line 79
Point 8: Line 126-128: According to [37], ??
Response 8: the author name was added, see line 80.
Point 9: Line 96: The ability of tomato plants to withstand WSS circumstances was investigated by evaluating some physio-biochemical indices "Remove".
Response 9: the paragraph was removed.
Point 10: Line 147-148: Forty-day-old tomato (Solanum lycopersicum L.) transplants (cultivar 320) were secured by the Nurseries of Agriculture Ministry, Cairo, Egypt " Needs rewriting"
Response10: Thanks for this valuble comment. The sentence was rewritten (Lines 549-550).
Point 11: Line 157-161: This concentration (2.5 g L−1) of the NABF was selected based on a preliminary study, in which five times foliar nourishment with 2.5 g NABF L−1 gave the best tomato growth results among several concentrations used (data not shown). Additionally, this formulation was selected from several formulations due to its best results (data not shown). The NABF composition is presented in Table 3. (It is required to clarify and review the data that is not displayed and add it to supplementary data).
Response11: The requested data to be provided as Suplementary Materials have been written as a manuscript and submitted to “Journal of Plant Growth Regulation”.
Point 12: This formulation was used at a rate of 2.5 g L−1. (Remove).
Response12: Done as requsted.
Point 13: Line 176: Also, the fertilization program detailed in [46] with some modifications. "Needs rewriting".
Response13: Done as requsted (Lines 582-583).
Point 14: Twelve weeks after transplantation (WAT), plant samples were taken. Randomly, 9 plants were taken from each treatment (3 plants from each row/replicate) for growth traits, photosynthetic efficiency parameters, and nutrient contents. In the same way, another 9 plants were taken and assigned to enzyme assays, enzyme gene expression, and hormonal analysis. A third set of 9 plants was taken randomly from each treatment and assigned to further analyses (leaf integrity parameters, oxidative stress markers, osmoprotectants (OPS), and non-enzymatic antioxidants). Beginning at 12 WAT, tomato fruits were collected five times, every seven days, from the remaining plants in each treatment. The fruits were used to evaluate the fruit yield components and quality of fruits. (The paragraph needs more explanation of analysis methods and references).
Response14: The fruit yield characteristics, and fruit quality traits, leaf integrity parameters, oxidative stress markers, osmoprotectants (OPS), and non-enzymatic antioxidants measurements and it referneces was displayed in:
2.5. Fruit Yield Characteristics, and Fruit Quality Traits.
2.9. Assessments of Osmoprotectants (OPS) and Antioxidant Level
2.10. Assay of Antioxidant Enzyme Activities and Enzyme Gene Expression).
Point 15: 2.5. Fruit Yield Characteristics, and Fruit Quality Traits:
Wrong title and what is mentioned after the title has nothing to do with the title. This indicates that the manuscript was not reviewed.
Response 15: the paragraph was changed. (Lines 622-624)
Point 16: NABF, Is it a commercial product?
Response 16: No, it is not a commercial product, but it is a formula that we have formulated and we are now working on the procedures to register it as a patent. Then we will conduct some other research on it on various other crops and other environmental stresses in preparation for its approval as a commercial product for the agricultural market.
Point 17: Results: The titles of all tables need to be revised, for example not mentioning the treatment 100%.
Reponse 17: Titles of all tables have been modified.
Point 18: Fig. 1,2 ,3,4. Need to improve.
Reponse 18: Improved.
Point 19: Discussion: a good portion will do. But it can be shortened.
Response 19: We would like to thank Reviewer #2 for this note and the discussion section are shortened
Point 20 :Conclusions: can be shortened.
Response 20: done as requsted (see the conclusion part)
Point 21: References: Scientific names are written in italics.
Reponse 21: done as requsted (see the references part).

Round 2
Reviewer 2 Report
The authors have addressed all of my queries. The revised version is significantly improved and can be accepted for publication.
Thanks
Author Response
Dear Reviewer# 2, based on your comment “The authors have addressed all of my queries. The revised version is significantly improved and can be accepted for publication”, we thank you very much and appreciate your efforts.
